# High-Quality Queens Produce High-Quality Offspring Queens

**DOI:** 10.3390/insects13050486

**Published:** 2022-05-23

**Authors:** Longtao Yu, Xinxin Shi, Xujiang He, Zhijiang Zeng, Weiyu Yan, Xiaobo Wu

**Affiliations:** Jiangxi Province Key Laboratory of Honeybee Biology and Beekeeping, Honeybee Research Institute, Jiangxi Agricultural University, Nanchang 330045, China; ylti0529@163.com (L.Y.); shixinxin0113@163.com (X.S.); hexujiang3@163.com (X.H.); bees1965@sina.com (Z.Z.); ywygood-0216@163.com (W.Y.)

**Keywords:** maternal effects, queen rearing, gene expression, queen quality, offspring

## Abstract

**Simple Summary:**

Maternal effects are a wise strategy for animals to adjust their offspring quality. Honey bee queens can adjust their investments in their female offspring to maximize the use of social resources, and these different investments may eventually lead to differing offspring quality. However, maternal effects on the quality of reared queens in artificial queen rearing has been ignored. To elucidate the maternal effects on the phenotypic and genetic alterations of reared queens, we examined the effect of ova sources caused by maternal effects on the development of reared queens and their offspring queens. Our results will shed light on improving the quality of reared queens and promoting the development of beekeeping.

**Abstract:**

Honey bees, rather than rear queens with eggs and larvae from worker cells, prefer to rear new queens with eggs form queen cells, if available. This may be a result of long-term evolutionary process for honey bee colonies. However, the exact mechanism of this phenomenon is unclear. In this study, queens were reared with eggs from queen cells (F1-QE), eggs from worker cells (F1-WE), and two-day-old larvae from worker cells (F1-2L). Physiological indexes and the expression of the development-related genes ((*Hexamerin* (*Hex*110, *Hex*70b), *Transferrin* (*Trf*), and *Vitellogenin* (*Vg*)) of reared F1 generation queens were measured and compared. Furthermore, F2 generation queens were reared with one-day-old larvae from F1 queens, and the weight and ovariole count of reared F2 generation daughter queens were examined. Meanwhile, the expression of the development- and reproduction-related genes (*Hex*110, *Hex*70b, *Trf*, *Vg*, and *Juvenile Hormone* (*Jh*)) and immune detoxication-related genes (*Hymenoptaecin*, *Abeacin*, and *Cyt*P450) of reared F2 queens were further explored. We found that the F1-QE queens had the highest physiological indexes and higher *Hex*110 and *Trf* expression levels, while no significant difference was found in the expression of *Hex*70b and *Vg* among the three groups of F1 queens. In addition, the reared queens of F2-QE had the highest quality, with the highest development, reproduction, immune-detoxication genes’ expression levels. Our results revealed that the quality of reared offspring queens from high-quality mother queens was also high. These findings inform methods for rearing high-quality queens and highlight that a high-quality queen is essential for offspring colony growth and survival.

## 1. Introduction

The honey bee (*Apis mellifera*) is a eusocial pollinator, with substantial importance to the ecosystem. A honey bee colony normally contains a single queen, and colony success depends highly on this one individual. The queen is normally responsible for all egg laying and brood production within the colony and is especially important for colony growth and survival [1].

Artificial honey bee queen rearing is vital in beekeeping to regularly expand colonies, to minimize swarming tendency, to enhance brood and honey production, to increase colony number, and to improve their genetic characteristics [2]. Beekeepers are appreciative of high-quality queens, as they ultimately lead to a greater production of bee products and more economic income. However, the population of honey bee colonies have dramatically declined in many countries [3]. Indeed, commercial beekeepers have frequently reported that queen failure has been considered as one of the most important factors leading to colony losses in the past several years [4]. Inappropriate use of artificial queen rearing techniques will lead to a decline in honey bee colonies [5]. Consequently, more researchers and beekeepers are focusing on how to artificially rear high-quality queens.

In apiculture, the egg-laying ability and reproductive potential of a queen represents its “quality”, which results from its genome, developmental and rearing conditions, mating success, and beekeeping methods [5,6,7,8,9,10]. High-quality virgin queens usually show the following characteristics: a higher length of queen cell [5], more remaining royal jelly in the queen cell [5], a higher weight [11,12,13], a larger thorax [6], larger ovaries [11,14], and more ovarioles of newly emerged queens [7,15]. Additionally, a high-quality mated queen has a strong egg-laying ability, with more stored sperm [7] and higher sperm viability [16]. Additionally, a high-quality virgin or mated queen produces more pheromones, which will benefit for colony growth and cohesion [6]. In contrast, lower pheromone production reduces labor and product production in honey bee colonies [1]. Moreover, researchers have revealed that queens with these high-quality traits usually have a higher expression level of development- and reproduction-related genes, including *Hexamerin* (*Hex*) [17,18], *Vitellogenin* (*Vg*) [19,20,21], *Transferrin* (*Trf*) [20,22,23], *Juvenile Hormone* (*JH*) [24,25,26], *Abeacin* [27,28], *Defesin* [28,29], *Hymenteaicn* [28,30], and *Cyt*P450 [31,32]. Hence, these traits could be considered for evaluating the quality of queens.

The queen presents one of the most impressive documented cases of developmental plasticity [33], while the evolutionary strategy for honey bees should be considered as a factor affecting the quality of queens in artificial queen rearing. During a long-term evolutionary process, honey bees evolved a series of strategies to better adapt to their environment. For instance, honey bee queens are highly polyandrous. Honey bee virgin queens take one or several mating flights to mate with many drones [7]. Their mating behavior usually occurs at a considerable distance from their native colonies. This behavior eliminates low-quality drones, avoids inbreeding, and ensures genetic diversity and offspring quality, which is beneficial for the reproductive quality of queens and honey bee colony growth [8]. Furthermore, the development of queens was influenced by many factors in artificially queen rearing, such as maternal effects [28,34] and the age of grafted larvae [9,15]. Maternal effects are the offspring phenotypes that are influenced by the mother rather than its own genotype, and they widely exist in animals and plants [35,36,37]. Parents can adjust their offspring quality to adapt to the environment, and it is an important mechanism of adaptive phenotypic plasticity [38]. For instance, ant queens (*Pheidole pallidula*) and honey bee queens (*Apis mellifera*) can selectively lay larger eggs, resulting in high-quality offspring [8,38,39,40,41]. Wei et al. reported that eggs in queen cells were much larger than eggs in worker cells, which may cause significant differences in the morphology of workers and queens [28]. Moreover, some evidence shows that rearing queens by grafting eggs compares favorably to doing so by grafting larvae, and that the quality of reared queens decreases as the larval age increases [5,9,15]. Hence, the environmental conditions can be changed to improve the quality of reared-queens.

Previous studies indicated that maternal effects cause differences in queen development, and the quality of queens reared with eggs from queen cells (QEs) is higher than that reared with WEs (eggs from worker cells) and 2L (two-day-old larvae in worker cells) [28,34]. However, the effect of maternal effects on the expression of development-related genes is unclear, and the effect of the differences caused by maternal effects on the quality of offspring queens is still ambiguous. Here, we reared queens with QE, WE, and 2L and compared the quality of F1 queens among these three groups, as well as the quality of F2 queens from each of the three groups.

## 2. Materials and Methods

### 2.1. Insects and F1 Generation Queens Rearing

Honey bee (*Apis mellifera*) colonies were kept at the Honey Bee Research Institute of Jiangxi Agricultural University (28.46° N, 115.49° E), according to standard beekeeping methods. All experiments were repeated three times by using three different mother queens and were conducted at three colonies. Experiments were conducted from April to June 2021. A natural mated queen was controlled for eggs laying in queen cells for 12 h, and the queen was controlled and allowed to lay eggs in an empty worker comb for 12 h. After that, about 64 eggs were laid in queen cells (QE), and 128 eggs were laid in worker cells (WE). These eggs in queen cells were transferred to a super box for hatching, and some eggs in worker cells were transferred to artificial queen cells and kept in a super box. The rest of worker-cell eggs were kept in their native colony until they hatched to two-day-old larvae (2L), and these two-day-old worker larvae were transferred to queen cells and kept in a super box. Queen cells of the QE, WE, and 2L groups were kept in the same honey bee colony. When the newly reared queens were about to emerge, these queen cells were placed in a dark incubator (34 °C, 75%) for obtaining the new queens.

The length of the queen cells was measured when they had been completely capped. About 20 queens of each group were collected for all experiments. The weight of the remaining royal jelly in the queen cells of eight newly emerged queens of each group, as well as their weight and the thorax width, were measured. The rest of the newly emerged queens were transferred to queen cages and placed in queen-less colonies for 5 days, where they could be fed and tended by workers through the cage. Five days later, 10 queens of each group were dissected to obtain the leftover ovaries for measuring the expression level of the development-related genes of queens.

### 2.2. Gene Expression Analysis

Each sample had an ovary from one queen, and each group contained 10 samples. The total RNA of queen ovary was extracted with the TransZol Up Plus RNA kit (TransGen Biotech, Beijing, China). The concentration and purity of the RNA were measured using a Nanodrop 2000 spectrophotometer (Thermo Scientific, Wilmington, DE, USA). RNA integrity was determined by running an aliquot on 1% agarose gel. RNA with high purity and integrity was used to synthesize cDNA by using a reverse transcription kit (Takara, Tokyo, Japan) and conducted using a PCR instrument (T100 Thermal Cycler, Bio-Rad, CA, USA). RNA and the reverse transcripts were preserved in a −80 °C freezer.

Primers are shown in Table 1 and were designed with Primer 5.0 software and synthesized at Sangon Biotech (Shanghai, China). The RT-qPCR reaction was accomplished with a PCR instrument (QuantStudio™ 5, Applied Biosystems, MA, USA) by using an RT-qPCR kit (SYBR^®^ Premix Ex Taq™ II, Takara, Tokyo, Japan). Each reaction had four technical replicates, and β-actin and GAPDH were treated as the internal reference gene.

### 2.3. F2 Generation Queens Rearing

According to the above method, queens of the F1 generation were reared with three kinds of ova sources and were allowed to mate naturally. When the queens lay eggs stably, we controlled F1 queens of QE, WE, and 2L for laying eggs in worker cells for about 2 h, respectively. We harvested about 30 eggs of each queen, and these eggs were kept in their native hive for hatching to 1-day-old larvae. These larvae were transferred to queen cells and kept in the same colony. Queen cells of F2-QE, F2-WE, and F2-2L were placed in a dark incubator (34 °C, 75%) when they were about to emerge.

### 2.4. Physiological Indexes of F2 Queens

Approximately 20 queens of each group were collected. The weight of newly emerged F2 queens was measured immediately after emerging from queen cells. About 15 queens were kept in a queen-less colony for 5 days. The abdomen of 5-day-old F2 queens was dissected, and the digestive tract and honey sac was removed. The ovaries from 10 different queens of each group were then harvested. The left ovaries were used for measuring the expression level of the development-related genes of F2 queens. The right ovaries were used for making the paraffin section, and ovarioles were counted. The methods of making the paraffin section and counting the ovarioles are described in [5,42].

### 2.5. Statistical Analysis

Statistical analyses were performed using SPSS Statistics version 26, and outliers (defined as the mean ± 3 times the standard error) were removed. All differences were determined by one-way ANOVA, and Fisher’s LSD tests were used to determine if there were any differences among different groups. The queens’ gene expression level was calculated using the $2^{-(\Delta \Delta Ct)}$ method [43].

## 3. Results

### 3.1. The Physiological Indexes of Reared F1 Queens

The lengths of the queen cells ($F_{2,22}=14.087,p<0.001$, Figure 1) of F1-QE and F1-WE were significantly higher than those of F1-2L, but the difference between F1-QE and F1-WE was not significant. When reared queens emerged from queen cells, the weight of the remaining royal jelly ($F_{2,33}=30.713,p<0.001$, Figure 1) in the queen cells of F1-QE was significantly higher than that of F1-WE and F1-2L, while there was no statistical significance between F1-WE and F1-2L. Moreover, the weight of the newly emerged queens of F1-QE were significantly higher than F1-WE, and that in F1-WE were significantly higher than in F1-2L ($F_{2,14}=25.307,p<0.001$, Table 2). The thorax width ($F_{2,17}=9.534,p<0.005$, Table 2) of queens of F1-QE and F1-WE were significantly higher than that of F1-2L, but there was no significant difference in the thorax width of queens between F1-QE and F1-WE.

### 3.2. The Expression of Development-Related Genes of F1 Queens

As shown in Figure 2, the relative expression of Hex110 and Trf in the queens of F1-QE was significantly higher than that of F1-WE and F1-2L, while that of F1-WE did not differ from F1-2L (*Hex*110: $F_{2,14}=7.076,p<0.05$; *Trf*: $F_{2,16}=11.258,p<0.05$). However, there was no significant difference in the expression of *Hex*70b and *Vg* among the three groups of queens (*Hex*70b: $F_{2,9}=0.125,p=0.884$; *Vg*: $F_{2,10}=0.527,p=0.606$).

### 3.3. The Weight and Ovarioles Number of F2 Queens

As shown in Table 3, the weight ($F_{2,10}=5.604,p<0.05$) of newly emerged F2-QE queens were significantly higher than that in F2-WE and F2-2L, while there was no significant difference between F2-WE and F2-2L. Meanwhile, the number of ovarioles ($F_{2,18}=3.254,p=0.062$) of the F2-QE queens was significantly higher than in F2-2L, but no significant difference was found between F2-QE and F2-WE or between F2-WE and F2-2L.

### 3.4. The Relative Expression of the Development-Related Genes of F2 Queens

The results are shown in Figure 3. Our study found that there were significant differences in the relative expression level of *Jh*-like ($F_{2,10}=7.742,$p<0.05), *Hex*110 ($F_{2,18}=13.332,$p<0.005), *Cyt*P450 ($F_{2,15}=3.782,p<0.05$), *Hymenoptaecin* ($F_{2,22}=8.842,p<0.005$), *Hex*70b ($F_{2,23}=3.939,p<0.05$), and *Trf* ($F_{2,22}=3.674,p<0.05$), and there was not a very significant difference in the relative expression level of *Vg* ($F_{2,23}=2.801,p=0.082$) and *Abeacin* ($F_{2,24}=2.902,p=0.074$). The relative expression levels of *Jh*-like, *Hex*110, *Cyt*P450, and *Hymenoptaecin* in F2-QE queens were significantly higher than those in F2-WE and F2-2L, but there was no statistical difference between F2-WE and F2-2L. Moreover, the relative expression levels of *Vg*, *Hex*70b, and *Abeacin* in F2-QE queens were significantly higher than those in F2-2L, but no significant difference was found between F2 -QE and F2-WE or between F2-WE and F2-2L. The relative expression levels of Trf in F2-QE queens were significantly higher than that in F2-WE, but no significant difference was found between F2-QE and F2-2L or between F2-WE and F2-2L.

## 4. Discussion

The honey bee is a highly developed and astute social insect with a clear social division of labor, and exhibits a high degree of adaptability to the environment [44]. The quality of a queen is closely related to the honey bee colony’s adaptability because the development and growth of a colony depends on its reproductive capacity [1]. In fact, during a long period of evolution, honey bee colonies have developed an intraspecific competition and a system of superiority and inferiority within the colony [45].

A new queen will be reared when the old queen’s reproductivity decreases or the honey bee is ready to swarm in a natural colony. At this time, worker bees will build some queen cells and allow the old queen to lay eggs in queen cells. These eggs in the queen cells will develop into new queens. Instead, worker bees prefer to use eggs in worker cells rather than larvae to rear new queens in emergency queen rearing when the honeybee colony loses its queen suddenly [1,46]. Researchers have conducted numerous studies accordingly and have found that eggs laid in queen cells are larger than eggs laid in worker cells, and queens reared with queen-cell eggs are larger than those reared with worker-cell eggs [28,34]. Furthermore, evidence has indicated that the quality of queens reared by worker-cell eggs is higher than those reared by worker-cell larvae [5,9,15]. It can be inferred that the quality of queens reared in natural queen cells is higher than queens reared in emergency queen cells. For this reason, worker bees prefer queen laying eggs in queen cells for rearing new queens, while new queens are reared with eggs in worker cells instead of larvae in emergency queen rearing. Worker bees prefer to rear high-quality queens, which are vital for their colony growth and survival [1]. Indeed, high-quality queens produce more pheromones, and the cohesion and development of a honey bee colony depend on their queens’ pheromone [1]. Furthermore, a high-quality queen has a higher reproductive capacity, so that their colony is strong for resisting external negative environments [7,15].

Rearing queens by grafted larvae is the most common method in commercial beekeeping. However, some beekeepers ignore the effect of larval age, nutrition, and developmental space on the quality of reared queens [5,10,47]. Previous studies have shown that maternal effects impact the development of reared queens [28,34]. Our results of F1 queens are consistent with those of previous studies. We found F1-QE queens had the highest weight and thorax width. In addition, the queen cell lengths of F1-QE and F1-WE were longer than those of F1-2L. Related studies have revealed that developmental space has an effect on larval development [5,47]. Queens of F1-QE had a larger developmental space, which is beneficial for larval development. As the developmental space increased, the quality of reared queens increased [5,47]. Queens of F1-QE stayed in the larger developmental space before emerging from queen cells. Queens of F1-WE remained in worker cells for the first two days and remained in queen cells in the following days. F1-2L larvae hatched in worker cells and developed in queen cells. Nurse bees supply larvae in queen cells with an abundance of royal jelly and provide larvae in worker cells with far less worker jelly or a mixture of pollen and nectar, resulting in large differences in morphology between queens and workers [10]. Food with different nutrient content has been reported to be one of the most important factors affecting larval development [10,48]. Our results showed that the residual royal jelly in F1-QE queen cells was the highest, while there was less in the F1-WE and F1-2L queen cells. This demonstrated that the developmental and nutritional environment of F1-QE queens differed from that of F1-WE and F1-2L. We propose that there is less time for nurse bees to provide larvae in worker cells, and that they have less time prior to pupation to consume the rich royal jelly diet with its distinctive nutrition [5]. In addition, we propose that QE larvae may produce more pheromones, and that nurse bees may perceive the pheromones of larvae and provide larvae in QE queen cells with more fresh royal jelly [49]. This may explain both the lowest weight of F1-2L queens and the less residual royal jelly, while F1-QE queens showed higher reproductive traits and more residual royal jelly. Worker bees provided QE larvae with more royal jelly and had sufficient food and larger developmental space, so the quality of queens of F1-QE was higher.

*Hexamerin* (*Hex*) and *Transferrin* (*Trf*) are two crucial types of protein in the honey bee [17,18,20,22,23]. Vitellogenin (*Vg*) is the precursor of yolk protein synthesis, which provides nutrients and functional substances for the development of insects [19,20,21]. Related studies have revealed that the expression level of *Hex*, *Trf*, and *Vg* in the activated ovaries of worker honey bees were upregulated. It has been suggested that Hex, Trf, and Vg are probably related to the development of ovaries in honey bee [17,18,19,20,21,22,23]. Remarkably, our results showed that the queens of F1-QE had a higher expression level of *Hex*110 and *Trf* genes, but there was no significance in the expression of *Hex*70b and *Vg* genes among the three groups of queens. It was possible that maternal effects had a great impact on the reproductive potential of honey bee queens, and that the genetic mechanisms were altered by ova sources, which eventually caused differences in the quality of reared queens. Interestingly, we found that the expression levels of *Hex*70b and *Vg* genes in F2-QE were significantly higher than those in F2-2L. We predicted that the differences in F1 queens would be amplified in their offspring. Strangely, the expression level of *JH* was surprisingly higher in the queens of F2-QE, and the expression level of the Trf gene was upregulated in the queens of F2-2L. *Juvenile hormone* (*JH*) is one of the most important insect hormones in the process of insect metamorphosis [24,25,26]. The reason for this phenomenon is unclear. We considered that this opposite trend was affected by the environment. The specific mechanisms need to be further investigated.

Moreover, related studies have revealed that many immune- and detoxication-related genes, including *Abaecin*, *Hymenoptaecin*, and *Cyt*P450 may involve in regulating the development of honey bees [27,28,29,30,31]. Compared to workers, the expression levels of immune- and detoxication-related genes were upregulated in queens, as well as in the activated ovaries of workers [31,32,50,51]. In our study, we found that the expression of *Abaecin*, *Hymenoptaecin*, and *Cyt*P450 in the queens of F2-QE were the highest, suggesting that the immune-detoxication of offspring was affected by maternal effects, and the immune-detoxication of queens of F2-QE was superior. We observed that the queens of F2-QE had the highest newly emerged weight and the highest number of ovarioles. In summary, the expression level of development-, immune-, and detoxication-related genes showed a decreasing trend among the three groups of F2 queens. Queens of F2-QE had higher expression levels and physiological indexes, contrary to the queens of F2-2L. This trend demonstrated that the quality of the offspring of high-quality queens was also high, but the quality of the offspring of low-quality queens was lower.

We can conclude that maternal effects are a result of evolution in honey bees. By maternal effects, the mother queen selectively produces new queens and workers. The different laying behavior changes the expression trends of crucial genes in queens, thus altering the phenotype of offspring queens, resulting in a different quality. Moreover, reared queens with eggs in queen cells as well as their offspring queens had a higher quality. Our findings provide a basis for rearing high-quality queens, but our methods of laying eggs in queen cells are too limited to expand in apicultural and agricultural production. We are constantly improving our methods to obtain a sufficient amount of queen-cell eggs. Furthermore, our results cannot yet represent all genetic changes in the overall developmental and reproductive functions of honey bee queens, and the effect of maternal effects on the reproductive potential of reared F2 generation daughter queens and the exact mechanism contributing to the differences in the developmental and reproductive traits of F2 queens are not well understood.

## 5. Conclusions

Maternal effects induced mother queen laying different eggs in queen cells and worker cells. The different ova sources had a profound influence on the development and reproductive potential of reared queens. Our results showed that F1-QE queens had the highest quality, and the quality of F2-QE queens were higher than the other two groups of F2 queens. The quality of the offspring of high-quality queens was also high. This showed that maternal effects changed the phenotype and genotype of reared queens, and these changes will transmit to their daughter queens.

## Figures and Tables

**Figure 1 insects-13-00486-f001:**
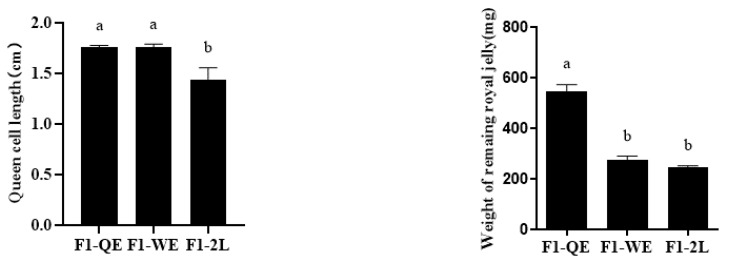
Queen cell length and weight of the remaining royal jelly in queen cells among the three groups. Bars show mean ± SE (standard error). Different letters above the bars indicate significant differences between groups (*p* < 0.05, one-way ANOVA test followed with Fisher’s LSD test).

**Figure 2 insects-13-00486-f002:**
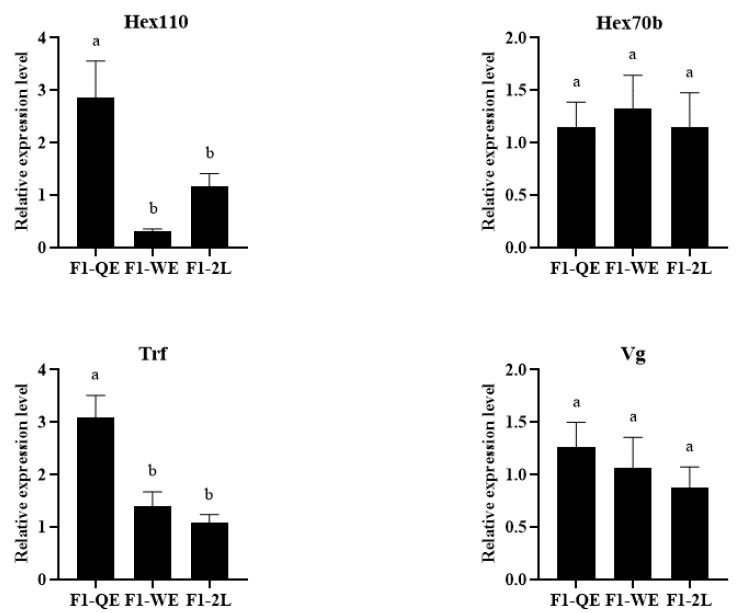
The expression level of development-related genes of F1 queens.Bars show mean ± SE (standard error). Different letters above the bars indicate significant differences between groups (*p* < 0.05, one-way ANOVA test followed with Fisher’s LSD test).

**Figure 3 insects-13-00486-f003:**
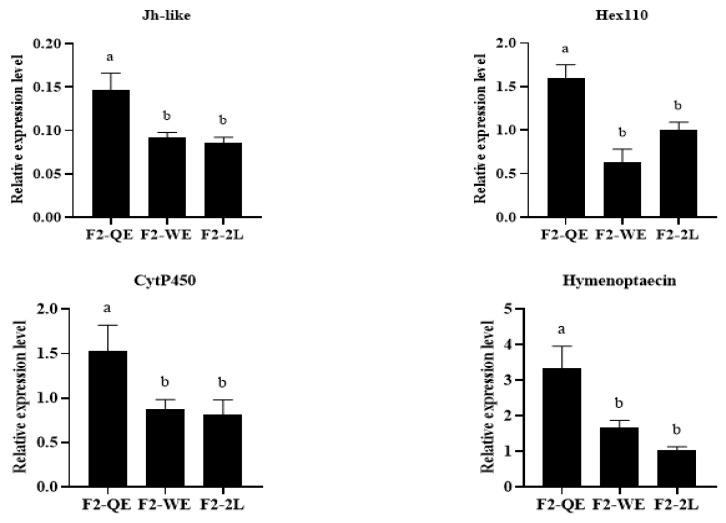
The expression level of development-related genes of reared F2 queens.Different letters above the bars indicate significant differences between groups (*p* < 0.05, one-way ANOVA test followed with Fisher’s LSD test).

**Table 1 insects-13-00486-t001:** Primers used in quantitative PCR and their sequences.

Target Gene	Forward Primer 5*′*-3*′*	Reverse Primer 5*′*-3*′*
*GAPDH*	GCTGGTTTCATCGATGGTTT	ACGATTTCGACCACCGTAAC
β-actin	TCCTGCTATGTATGTCGC	AGTTGCCATTTCCTGTTC
*Hex*110	AACGTGCCAGGCGCAGTTGT	TTCACCAGCATGGAGGTTCTGGA
*Hex*70b	GAGGACGGTAGCGAGTCCTT	ATGTTGCGGCCCAATACAGG
*Trf*	AGCGGCATACTCCAGGGAC	CGTTGAGCCTGATCCATACGA
*Jh*-like	CACTGGCACCAGAGCCTGTC	GATTCCCATTGAACGAGCGA
*Abeacin*	TCTTCGCACTACTCGCCACG	TCAGGGACCATTCAATCCGA
CytP450	CAAAATGGTGTTCTCCTTACCG	ATGGCAACCCATCACTGC
*Hymenoptaecin*	TCAAGCGGAATTGGAACCTG	CGAATCCTGCATGCTGTCGA
*Vg*	CGTGTTCCAGAGGACGTTGA	ACGCTCCTCAGGCTCAACTC

**Table 2 insects-13-00486-t002:** Weight and thorax width of F1 queens.

Groups	Weight of Newly Emerged Queens	Thorax Width
F1-QE	267.50 ± 6.33 a	4.94 ± 0.07 a
F1-WE	235.40 ± 7.75 b	4.88 ± 0.06 a
F1-2L	177.28 ± 12.41 c	4.66 ± 0.067 b

Data in the table are mean ± SE (standard error). a,b,c are different letters used to represent significant differences (*p* < 0.05, Fisher’s LSD test) following the data in the same column.

**Table 3 insects-13-00486-t003:** The newly emerged weight and the ovariole count of F2 queens.

Groups	Weight of Newly Emerged Queens	Number of Ovarioles
F2-QE	277.30 ± 6.33 a	116.71 ± 4.70 a
F2-WE	257.53 ± 6.16 b	107.71 ± 3.60 ab
F2-2L	262.88 ± 1.98 b	103.43 ± 2.71 b

Data in the table are mean ± SE (standard error). a,b are different letters used to represent significant differences (*p* < 0.05, Fisher’s LSD test) following the data in the same column.

## Data Availability

The data presented in this study are available in this article.

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
