# Peer review of "High-Quality Queens Produce High-Quality Offspring Queens"

_insects, 2022, doi:10.3390/insects13050486_

Round 1
Reviewer 1 Report
The current paper compared the impact on morphology, physiology and gene expression of queen reared in different condition after emerging. The paper is very interesting and the experiment well designed.
I still have comment, that are mainly specific points to precise:
Did the authors used both or one queen ovary for gene expression of F1 queens?
*
Introduction
P2 second paragraph describe the characteristics of high quality queen, the authors didn’t mentioned pheromone production (differences between virgin or mated queens OR even between mated queens pheromone production is different according to mating history.)
Material and method
By reading the material, it sounds that all the experiment was realized on 1 colony. Could the author confirmed. As the beginning of the paper start with a plural, I’m not 100% sure.
Is the queen of this colony naturally mated? (And so mated with several males?)
The authors mentioned here 64 eggs (QE) and 128 WE, but later in the paper and in the results section, the authors could also mentioned the number of replicate. Did they all emerged? How many were used for the statistical analysis? How many were dissected and tested for gene expression?
Discussion,
Sentence starting by “furthermore…” specify by eggs if its referring to eggs in worker-cell ?
Sentence “it was suggested…”, could the authors add a reference for the statement of high quality queens. In particular, which study demonstrate that the different rearing condition lead to different queen quality.
The conclusion of the paragraph “rearing queen”. The authors conclude that F1 QE have more residual royal Jelly that the other that could affect queen quality. But in the 3 groups cells have left over royal jelly meaning that the food was not a limitant factor and they all have sufficient food. So I don’t understand why this variable could be link with queen quality for F1 QE? The significant value between different groups didn’t necessary means that there is also significant consequences on individuals. Could the authors better explain their conclusion in this regard?
The last sentence of the paragraphe starting by “moreover…”
I’m still uncomfortable on how the authors linked their variables to queen quality and offspring quality. Could the authors be more specific or better define how they used terme quality? Beside gene expression differences. The authors could also add references on paper making the link between differences in gene expression and “quality”?
Author Response
Response to Reviewer 1 Comments
The current paper compared the impact on morphology, physiology and gene expression of queen reared in different condition after emerging. The paper is very interesting and the experiment well designed.
Response: We appreciate the reviewer for recognizing the significance of this work and for the insightful comments, which substantially improve the quality of this paper.
I still have comment, that are mainly specific points to precise: Did the authors used both or one queen ovary for gene expression of F1 queens?
Response: Ten queens of each group were dissected to get the ovary for measuring the expression level of development-related genes of queens. Each sample had an ovary from one queen, and each group contains ten samples. Ten ovaries were obtained for gene expression of F1 queens and F2 queens.
Introduction
P2 second paragraph describe the characteristics of high quality queen, the authors didn’t mentioned pheromone production (differences between virgin or mated queens OR even between mated queens pheromone production is different according to mating history.)
Response: Thanks for the valuable comment. Indeed, a high-quality queen with higher pheromone production is crucial for colony growth and cohesion. We had added this trait into the manuscript, as shown in Line44-53.
Material and method
By reading the material, it sounds that all the experiment was realized on 1 colony. Could the author confirmed. As the beginning of the paper start with a plural, I’m not 100% sure.
Response: All experiments were repeated three times by using three different mother queens and conducted at three colonies. As shown in the manuscript.
Is the queen of this colony naturally mated? (And so mated with several males?)
Response: Honey bee queens are highly polyandrous. In natural honey bee colonies, virgin queens take one or several mating flight to mate with many drones. Although the number of males for mating with queens should be considered as a key factor contributing to the difference in reproductive potential of queens. However, it is difficult to control the drone number for mating with queen. In addition, weight of newly emerged queens has been proved to correlate with the number of drones for mating with queens (Amiri et al.,2017). A larger queen may have a larger spermathecae and more sperm (Anita et al.,2012). In the present study, reared F1 queens were allowed to mate naturally and this study was aiming to explore the effect of different ova sources on the quality of reared F2 queens. We supposed that the number of males for mating with queens is determined by the queens themselves. All queens were kept at the same conditions, and F1-QE queens had the highest quality, and the quality of F2-QE were also higher than the other two groups of F2 queens, suggesting that maternal effects directly influence queen quality and the effects can be transmitted to later generations. In our following studies, multi-male fertilized queens will used to reduce the inlfuence of different mating number of drones.
The authors mentioned here 64 eggs (QE) and 128 WE, but later in the paper and in the results section, the authors could also mentioned the number of replicate. Did they all emerged? How many were used for the statistical analysis? How many were dissected and tested for gene expression?
Response: Approximately 20 queens of each group were collected, and 10 queens were used for measuring weight and thorax size, the remaining queens were dissected to get ovaries for gene expression and scoring the number of ovarioles. As shown in the manuscript.
Discussion,
Sentence starting by “furthermore…” specify by eggs if its referring to eggs in worker-cell ?
Response: Thanks for the comment, the “eggs” is “eggs in worker cells”. Here, we have added “worker-cell” before “eggs”. As shown in the manuscript.
Sentence “it was suggested…”, could the authors add a reference for the statement of high quality queens. In particular, which study demonstrate that the different rearing condition lead to different queen quality.
Response: Done! Thank you! As shown in the manuscript.
The conclusion of the paragraph “rearing queen”. The authors conclude that F1 QE have more residual royal Jelly that the other that could affect queen quality. But in the 3 groups cells have left over royal jelly meaning that the food was not a limitant factor and they all have sufficient food. So I don’t understand why this variable could be link with queen quality for F1 QE?
Response: Queens of F1-QE stayed at a larger developmental space before emerging from queen cells. While queens of F1-WE stayed at worker cells for the first 2 days and stayed at queen cell for the next days, and F1-2L larvae hatched in worker cells and developed in queen cells. Nurse bees supply larvae in queen cells with an abundance of royal jelly, and provide larvae in worker cells with far less worker jelly or a mixture of pollen and nectar, resulting in large differences in morphology between queens and workers (Shi et al.,2011). Additionally, Food with different nutrient contents has been reported to be one of the most important factors affecting larval development (Wang et al.,2014). We propose that there is less time for nurse bees to provide with larvae in worker cells, and they have less time prior to pupation to consume the rich royal jelly diet with its distinctive nutrition (Yi et al.,2020). In addition, we propose QE larvae may produce more pheromone, and nurse bees may perceive the pheromone of larvae and provide larvae in QE queen cells with more fresh royal jelly (Sagili et al.,2009). It is known that fresh royal jelly contains more active substances, which will be more conducive to the development of larvae. This may explain both the lowest weight of newly emerged queens and residual royal jelly, while F1-QE queens showed higher reproductive traits and more residual royal jelly.
A high-quality queen had a higher weight and more ovarioles, with more residual royal jelly in queen cells after emerging from queen cells (Yi et al.,2020). Hence, we propose the residual royal jelly in queen cells can be considered as a key trait for evaluating queen quality.
The significant value between different groups didn’t necessary means that there is also significant consequences on individuals. Could the authors better explain their conclusion in this regard?
Response: Thanks for the suggestion. Our results of the present study showed a significant difference among the three group of queens. But it doesn’t means that there is also significant consequences on individuals. Our results showed a trend that QE queens had the highest quality, and rearing queens with eggs from queen cells may better than that with eggs or larvae from worker cells. Meanwhile, our study is dedicated to improving the quality of queen, and we are working hard to promote the application in agricultural production. Therefore, our results will be more beneficial to many honey bee colonies than just a single queen. We attach great importance to your suggestion, and we have described the limitations of my methods in the discussion section. Thank you!
The last sentence of the paragraphe starting by “moreover…”
I’m still uncomfortable on how the authors linked their variables to queen quality and offspring quality. Could the authors be more specific or better define how they used terme quality? Beside gene expression differences. The authors could also add references on paper making the link between differences in gene expression and “quality”?
Response: As shown in Line44-58: In apiculture, the egg-laying ability and reproductive potential of a queen represents her “quality”, which results from her genome, developmental and rearing conditions, mating success, and beekeeping methods. High-quality virgin queens usually show the following characteristics: higher length of queen cell, more amount of remaining royal jelly in queen cell, higher weight, bigger thorax size and ovary size, more ovarioles of newly emerged queen. Also, a high-quality mated queens have a strong egg-laying ability, with more stored sperm and higher sperm viability. Additionally, a high-quality virgin or mated queen produces more pheromone, which will benefit for colony growth and cohesion. In contrast, lower pheromone production reduces labor and products production in honey bee colonies. Moreover, researchers reveled that the queens with these high-quality traits had a higher expression level of development and reproduction related genes, including Hexamerin (Hex), Vitellogenin (Vg), Transferrin (Trf), Juvenile Hormone (JH), Abeacin, Defesin, Hymenteaicn CytP450. Hence, these traits could be considered for evaluating the quality of queens.
Reviewer 2 Report
Abstract: First sentence: Honey bees instead of “Honey bee”. In addition, you need to mention that honey bees prefer to rear new queens from egg in the queen cells if available rather than rear queens from eggs and larvae from worker cells.
Introduction:
First paragraph; references 1-2 are cited for the wrong statement.
You need to define the “eugenics”, not giving some examples.
“ant queens (Pheidole pallidula) and honey bee queens (Apis mellifera) can selectively lay larger eggs, resulting in high-quality offspring”, you may need to cite Amiri et al 2020 (https://doi.org/10.1111/jeb.13589).
2.1. Material; since you explained everything in the following text I do not think you need a subsection for materials.
2.3. Gene expression analysis. This section needs to be rewritten, since there are many grammatical mistakes
In a FREZEER at -80 °C.
The authors did not mention how many queens they investigated for the F1 experiment and F2 experiment. Without knowing how many samples were tested it is difficult to judge the obtained results.
Author Response
Response to Reviewer 2 Comments
We appreciate the reviewer for insightful comments, which substantially improve the quality of this paper.
Abstract: First sentence: Honey bees instead of “Honey bee”. In addition, you need to mention that honey bees prefer to rear new queens from egg in the queen cells if available rather than rear queens from eggs and larvae from worker cells.
Response: Done! Thank you! As shown in the manuscript.
Introduction:
First paragraph; references 1-2 are cited for the wrong statement.
Response: Done! Thank you! As shown in the manuscript: The queen is normally responsible for all egg laying and brood production within the colony, her quality is especially important for colony growth and survival (Winston,1987).
You need to define the “eugenics”, not giving some examples.
Response: We attach great importance to your suggestion, “eugenics” is specific to human. Here, we have advised “eugenics” through the manuscript. Honey bees prefer to rear new queens with eggs form queen cells if available rather than rear queens with eggs and larvae from worker cells. This may be a result of long-term evolutionary process for honey bee colonies. In addition, during a long-term evolutionary process, honey bees evolved a series of strategies to better adapt to their environment. For instances, honey bee queens are highly polyandrous, honey bee virgin queens take one or several mating flight to mate with many drones. Their mating behavior usually occurs at a considerable distance from their native colonies place.
“ant queens (Pheidole pallidula) and honey bee queens (Apis mellifera) can selectively lay larger eggs, resulting in high-quality offspring”, you may need to cite Amiri et al 2020 (https://doi.org/10.1111/jeb.13589).
Response: Done! Thank you!
2.1. Material; since you explained everything in the following text I do not think you need a subsection for materials.
Response: Done! Thank you!
2.3. Gene expression analysis. This section needs to be rewritten, since there are many grammatical mistakes In a FREZEER at -80 °C.
Response: Done! Thank you! As shown in Line114-126: Each sample had an ovary from one queen, and each group contains ten samples. Total RNA of queen ovary was extracted by TransZol Up Plus RNA kit (TransGen Biotech, China). The concentration and purity of RNA were measured using a Nanodrop 2000 spectrophotometer (Thermo Scientific, Wilmington, DE, USA). RNA integrity was determined by running an aliquot on 1% agarose gel. RNA with high purity and integrity was used to synthesize cDNA by using Reverse transcription kit (SYBR® Premix Ex Taq™ II) and conducted at PCR instrument (T100 Thermal Cycler, Bio-Rad, USA). RNA and the reverse transcripts were preserved in a -80 ℃ freezer.
Primers was shown as Table 1, and were designed with Primer 5.0 software and synthesized at Sangon Biotech (Shanghai, China). The RT-qPCR reaction were accomplished an PCR instrument (QuantStudio™ 5, Applied Biosystems, USA) by using RT-qPCR kit (SYBR® Premix Ex Taq™ II). Each reaction had four technical replicates, β-actin and GAPDH were treated as the internal reference gene.
The authors did not mention how many queens they investigated for the F1 experiment and F2 experiment. Without knowing how many samples were tested it is difficult to judge the obtained results.
Response: Done! Thank you! As shown in this manuscript.
Reviewer 3 Report
Dear Authors,
The manuscript elucidates strategies commonly adopted within beekeeping that can effectively improve the quality of queen production. However, some points need further explanation as follows:
1) It is clear that only one queen was used to form all the data presented. I understand this suitability; however, so that the discussion can be more comprehensive, I suggest that you discuss the limitation of this choice since it is known that different families can have different gene expressions levels for the same genes;
2) In a commercial context, it is known that the rearing of queens using larvae is common. This research somehow validates the possibility of tremendous success with the creation of queens via eggs. Can these results be generalized or applied to any queen rearing system? What are the precautions for using these results in the field since the study is limited to a family that gives rise to eggs and larvae, and it is still the colony that reared the queens? The statistical design and methods meet the objective, but if its discussion permeates practical issues, the article can achieve a more significant impact, especially in the world queen breeding industry.
Author Response
Response to Reviewer 3 Comments
Dear Authors,
The manuscript elucidates strategies commonly adopted within beekeeping that can effectively improve the quality of queen production. However, some points need further explanation as follows:
Response: We appreciate the reviewer for insightful comments, which substantially improve the quality of this paper. We have taken all of the comments into consideration and further addressed the comments. The replies have been made one by one according to the comments made:
1) It is clear that only one queen was used to form all the data presented. I understand this suitability; however, so that the discussion can be more comprehensive, I suggest that you discuss the limitation of this choice since it is known that different families can have different gene expressions levels for the same genes;
Response: All experiments were repeated three times by using three different mother queens and conducted at three colonies. We realize that different genes in the same families may have different expression levels, and the changes of these genes’ expression levels aren’t comprehensive for all genes’ changes. We have added this part into the discussion section. And we attach great importance to your suggestion, in subsequent studies, we will also explore the effects of maternal effect on the honey bee queen entire genes through the transcriptome, as well as more comprehensive gene expression level involving other aspects genes of honey bee queen. Thank you!
2) In a commercial context, it is known that the rearing of queens using larvae is common. This research somehow validates the possibility of tremendous success with the creation of queens via eggs. Can these results be generalized or applied to any queen rearing system? What are the precautions for using these results in the field since the study is limited to a family that gives rise to eggs and larvae, and it is still the colony that reared the queens? The statistical design and methods meet the objective, but if its discussion permeates practical issues, the article can achieve a more significant impact, especially in the world queen breeding industry.
Response: We appreciate the reviewer for recognizing the significance of this work and for the insightful comments, which substantially improve the quality of this manuscript. Honey bees show dramatic developmental plasticity by creating queen cells into which eggs or young larvae from worker cells are transplanted. However, many studies revealed that the quality of reared queens by grafted larvae were lower than grafted eggs (Woyke et al.,1971; Yi et al.,2020). It is very difficult to graft eggs and thus grafting young larvae is a common approach in commercial queen rearing. For this reason, we invented an instrument for obtaining eggs in worker cells easily and rearing queens without grafting larvae (Pan et al.,2013; Wu et al.,2014). At present, we have further invented an instrument for obtaining eggs in queen cells (Wei et al.,2019; He et al., Cultivation device and method for queen bees: 201710855463.X). We found that the quality of queens reared by queen-cell eggs was better than worker-cell eggs. However, our methods of laying eggs in queen cells are too limited to expand in apicultural and agricultural production. Here, we are constantly improving our methods to get enough queen-cell eggs. We attach great importance to your suggestion, and we have described the limitations of my methods in the discussion section. Thank you!